# Understanding Tendon Fibroblast Biology and Heterogeneity

**DOI:** 10.3390/biomedicines12040859

**Published:** 2024-04-12

**Authors:** Sarah E. DiIorio, Bill Young, Jennifer B. Parker, Michelle F. Griffin, Michael T. Longaker

**Affiliations:** 1Hagey Laboratory for Pediatric Regenerative Medicine, Division of Plastic and Reconstructive Surgery, Department of Surgery, Stanford University School of Medicine, Stanford, CA 94305, USA; diiorios@stanford.edu (S.E.D.); younbil@stanford.edu (B.Y.); jparker6@stanford.edu (J.B.P.); mgriff12@stanford.edu (M.F.G.); 2Institute for Stem Cell Biology and Regenerative Medicine, Stanford University School of Medicine, Stanford, CA 94305, USA

**Keywords:** tendon, fibroblast, tenocyte, fibrosis, biomaterials

## Abstract

Tendon regeneration has emerged as an area of interest due to the challenging healing process of avascular tendon tissue. During tendon healing after injury, the formation of a fibrous scar can limit tendon strength and lead to subsequent complications. The specific biological mechanisms that cause fibrosis across different cellular subtypes within the tendon and across different tendons in the body continue to remain unknown. Herein, we review the current understanding of tendon healing, fibrosis mechanisms, and future directions for treatments. We summarize recent research on the role of fibroblasts throughout tendon healing and describe the functional and cellular heterogeneity of fibroblasts and tendons. The review notes gaps in tendon fibrosis research, with a focus on characterizing distinct fibroblast subpopulations in the tendon. We highlight new techniques in the field that can be used to enhance our understanding of complex tendon pathologies such as fibrosis. Finally, we explore bioengineering tools for tendon regeneration and discuss future areas for innovation. Exploring the heterogeneity of tendon fibroblasts on the cellular level can inform therapeutic strategies for addressing tendon fibrosis and ultimately reduce its clinical burden.

## 1. Introduction

Tendons attach muscles to bone and play a crucial role in the movement of the skeleton. They are present both in the case of synovial joints, such as the flexor tendons of the hands and feet, and in non-synovial joints, such as the Achilles, rotator cuff, and patellar tendons [1]. This anatomical difference allows for tendons to be classified as either synovial or non-synovial tendons (Table 1). Structurally, tendons comprise dense connective tissue with an extracellular matrix (ECM) predominantly made of type I collagen (collagen I). The ECM also includes proteoglycans, elastin, and glycoproteins [2]. The primary cells in the tendon are called tenocytes, along with the tendon stem and progenitor cells, which reside in the tendon sheath, specifically in the epitenon layer [3]. Together, these make up 90–95% of all cells in the tendon, with the remaining cells being vascular cells, synovial cells, chondrocytes near the insertion, and smooth muscle cells [4].

Although able to withstand significant forces, tendons are often subject to chronic overuse or acute tear injuries. The three most commonly injured tendons include the flexor tendons of the hand, the Achilles tendon, and the tendons of the rotator cuff [5]. Together, tendon injuries account for approximately 50% of the musculoskeletal disease burden, thus presenting an important pathology for study [6,7]. Tendon injuries can range from tendinopathy, which is defined as tendon degeneration often believed to be caused by microtears, to partial or complete tears [8,9,10]. Moreover, the tendon is a relatively avascular tissue and has a low cell density, leading to a decreased regenerative potential [11]. Oftentimes, tendons heal with an excess of disorganized ECM deposition, also known as fibrosis. Fibrotic scars can be problematic for patients, as they cause pain and restrict joint range of motion. Fibrosis can even progress to the point of contractures, where the tendon–muscle unit is actually shortened, not by muscle contraction, but by fibrosis [12].

**Table 1 biomedicines-12-00859-t001:** Summary of types of tendons with examples.

Type of Tendon	Definition/Function	Key Examples
Synovial	Surrounded by synovial sheath and bathed in synovial fluid	Flexor tendons of hands and feet
Non-synovial	Surrounded by paratenon	Achilles, rotator cuff tendons
Energy-storing	Provide elasticity	Achilles, patellar
Positional	Remain stiff to transfer force to bone	Supraspinatus

There are two main classification systems for tendons: synovial or non-synovial and energy-storing or positional. Definitions and examples are listed for each. This table was summarized from Kaya et al. [13].

Little is currently known about the specific cell subtypes and mechanisms that contribute to this fibrotic phenotype, and why some tendons form fibrosis or adhesions and others do not. However, increasing work is being conducted to understand tendon fibrosis and the roles played by various cell types, using techniques such as next-generation sequencing and lineage tracing. Several bioengineering solutions are also being developed to aid in tendon regeneration or prevent fibrosis, but continued innovation in this area is needed. In this review, we will discuss current knowledge of tendon fibrosis through the lens of tenocyte and fibroblast heterogeneity. We conducted this review using PubMed with search terms “tendon fibrosis”, “tendon heterogeneity”, “skin fibrosis”, and mainly focused on papers published in the last five years.

## 2. Overview of Tendon Healing: Cells and Signals

As previously discussed, tendons are a common source of musculoskeletal injury. It has been well described that after injury, tendons undergo three overlapping stages of healing: inflammatory, fibroblastic/proliferative, and remodeling (Figure 1) [14,15]. These stages mirror the stages of wound healing in the skin [16]. The inflammatory phase typically lasts from days 1 to 7 and involves the migration of immune cells, erythrocytes, platelets, and fibroblasts into the injury (Table 2). Neutrophil phagocytosis of necrotic tissue occurs in the first 24 h, followed by a mounting inflammatory response and continued phagocytosis by macrophages [8,17,18,19,20]. The fibroblastic, or proliferative, phase typically occurs between days 3 and 14 and involves the proliferation of fibroblasts and production of collagen III to provide a scaffold for migrating cells and new tissue formation [14,21]. During the remodeling phase, which begins around day 10 and can last for months or years post-injury, collagen III from the proliferative phase will be replaced by collagen I. Cross-linking of collagen fibers also occurs, increasing strength and elasticity of the tendon matrix [8].

Several growth factors play a role in each phase of tendon healing, and a single growth factor’s activity often continues through multiple phases (Table 2). In the inflammatory phase, platelets clot the blood vessel injury site immediately following injury and release platelet-derived growth factor (PDGF) and insulin-like growth factor I and II (IGF-I and IGF-II). PDGF plays its most significant role in this early stage by inducing the expression of additional growth factors and recruiting macrophages [22]. Macrophages and neutrophils, which migrate into the tissue, also release a cocktail of cytokines. Leukocytes, which migrate in, release basic fibroblast growth factor (bFGF), promoting cellular proliferation and migration, angiogenesis, and the excretion of other factors during all three phases [24,26,27]. During the proliferative phase, vascular endothelial growth factor (VEGF) promotes angiogenesis, and IGF-I promotes proliferation and migration of fibroblasts as well as the proliferation and differentiation of tendon stem cells [23,24]. The remodeling phase sees a continuation of VEGF expression, as well as PDGF and IGF-I to promote additional collagen production and the replacement of collagen III with collagen I [27].

One of the most important growth factors in tendon healing is transforming growth factor-beta (TGF-β), which plays an important role in every stage [25,27]. In the proliferative phase specifically, TGF-β promotes the new expression of collagen III and collagen I. TGF-β is also responsible for inducing expression of tenogenic markers such as *Scleraxis* (*Scx*) [24]. Growth and differentiation factors 5 and 6 (GDF-5 and GDF-6), which are members of the TGF-β superfamily, have also been shown to contribute to tendon healing as well [28,29]. Despite this healing process, the tissue formed in adults often contains fibrotic tissue, which does not have the same structural and mechanical properties as the original tendon. TGF-β in particular promotes fibrosis by upregulating collagen expression in the proliferative phase, as fibrous scar tissue is made up of disorganized collagen [25].

Matrix metalloproteinases (MMPs) are a family of enzymes which are another important factor in all phases of tendon healing [7]. In normal tendon development and maintenance, MMPs process and turn over collagen and promote new fibril growth and assembly. After injury, some MMPs degrade collagens, while others degrade glycoproteins and proteoglycans present in the tendons [15]. This degradation creates space for new, healthy tendon components to be produced and remodeled [15]. However, long-term activation of MMPs can be detrimental, because MMPs can create a low level of consistent inflammation, which can weaken tendons and contribute to tendinopathy or rupture [30,31].

## 3. Intrinsic and Extrinsic Cell Types in Tendon Healing

Although tendon healing is often described in terms of the fibroblast response, both extrinsic fibroblasts, which migrate into the injury and intrinsic tendon cells, called tenocytes, are believed to contribute to tendon healing [19,32,33].

Fibroblasts are generally defined as the cells which contribute to ECM production. While they lack a unified marker which describes the cell type, markers such as *platelet-derived growth factor receptor alpha* (*PDGFRα*), *vimentin*, and *CD34* have become associated with fibroblasts [34]. Activated fibroblasts, called myofibroblasts, are typically identified by the expression of *alpha smooth muscle actin* (*αSMA*). Fibroblasts are also sometimes defined by their lack of markers that are indicative of another cell type [35]. Anatomically, fibroblasts, which contribute to tendon healing, are located in the synovium and sheath of the tendon [36].

Tenocytes are also known as tendon fibroblasts and share multiple expression markers with extrinsic fibroblasts such as *collagen I*, *PDGFRα*, and *decorin* [37,38,39]. Like fibroblasts, there are no specific characterizing markers for tenocytes. However, *Tenomodulin* (*Tnmd*) is considered a typical marker for the cell type, while *Scx* is used as a marker of some tenocytes and tendon progenitor cell populations [38,40]. Anatomically, the tenocytes make up the tendon itself, as well as the endotenon and epitenon, which hold the tendon fiber bundles together and contain progenitor cells.

Flexor tendon explants cultured in vitro demonstrate intrinsic healing capacity, as illustrated by an increase in DNA and hydroxyproline content, representing an increase in tenocytes and collagen, respectively [41]. In the setting of tendon injury, however, this intrinsic regenerative capacity is often outweighed by the extrinsic fibroblasts, leading to collagen deposition and scarring [32,33,36]. The scar or adhesion that is formed prohibits smooth movement between the tendon and its surrounding synovium and sheath, leading to impaired motion.

## 4. Heterogeneity in Fibroblast Function

### 4.1. Fibroblast Heterogeneity in Skin

There is growing evidence that fibroblasts are not one unified cell population but that there is much heterogeneity between different tissues and even within each tissue [34,42,43]. Because fibroblast heterogeneity has so far been best characterized in skin, skin fibroblasts will be discussed first. There are several ways skin fibroblasts are classified, for example, dermal fibroblasts are often defined as either reticular or papillary [44]. This classification is determined partly by the anatomical location, as well as the cell morphology. Reticular fibroblasts are typically more square shaped in contrast to papillary fibroblasts, which are more spindle shaped [45]. There are also differences in cell expression: reticular cells contain more *αSMA*-positive fibroblasts and express genes associated with cytoskeletal organization and cell motility. Papillary fibroblasts, in contrast, are more associated with immune system response genes [45]. Another recent discovery found that there are subpopulations of fibroblasts that have distinctly different effects on wound healing; for example, in the dorsal skin, paraxial mesoderm-derived *Engrailed-1* lineage negative cells have been showed to contribute to regenerative, scarless healing, while *Engrailed-1* lineage positive cells contribute to fibrosis [46,47].

Despite the work that has already been performed to classify fibroblast subpopulations, there remains a need for further classification of subtypes. Recent single-cell RNA sequencing (scRNA-seq) work in skin has also challenged existing classification systems by finding that traditional markers of both dermal reticular and papillary fibroblasts were expressed across clusters or had very low expression. Instead, dermal fibroblasts were clustered into six groups based on gene ontology (GO)-term analysis [44]. Continued study of fibroblast populations, including those previously defined, will bring us closer to a complete understanding of heterogeneity in skin and other tissues.

### 4.2. Fibroblast Heterogeneity in Tendons

Compared to skin, there are relatively few studies which aim to identify fibroblast subpopulations in tendons. The studies that have been conducted mostly focus on the “tendon fibroblasts”, or tenocytes, and potential subpopulations of this cell type. Different studies have shown different numbers of subpopulations based on how data were clustered, tissue type (mouse or human), and whether injured or uninjured tissue was used [3,48,49,50,51,52]. Moreover, the terms tenocyte and tendon fibroblast were often used interchangeably in existing studies, and extrinsic fibroblasts were not always differentiated from tendon fibroblasts. Thus, the language used in each study will be maintained in this review.

By performing scRNA-seq on uninjured mouse Achilles tendon tissue, Micheli et al. identified three fibroblast populations classified by their high expression of collagen I component *Col1a1*. Subpopulations were named fibroblast 1, fibroblast 2, and junctional fibroblasts, differentiated by the expression of *osteopontin*, *dermatopontin*, and *type XXII collagen*, respectively [48]. In addition to these populations, the expression of typical tendon fibroblast markers such as *Scx*, *Tnmd*, and *Mohawk* (*Mkx*) was evaluated for each of the subpopulations. These markers were found to be present in some, but not all, of the fibroblasts in the fibroblast 1 and 2 populations. Further lineage tracing studies showed that *Scx* was present in developing tendons up to 2 months but was not ubiquitously expressed in adult tendon fibroblasts. This study further reinforced the lack of a unifying tendon fibroblast marker but determined some subpopulations worth further investigation.

Kendal et al. performed scRNA-seq and cellular indexing of transcriptomes and epitopes sequencing (CITE-seq) on human tendons from both normal and tendinopathy sources [49]. They identified five tenocyte subpopulations (Figure 2). All subpopulations shared a high expression of *COL1A1* and *COL1A2* and were defined by specific differentially expressed genes: microfibril associated (*KRT7/SCX+*), pro-inflammatory (*PTX3+*), pro-fibroadipogenic progenitors (*APOD+*), chondrogenic cells (*TPPP3/PRG4+*), and smooth muscle-mesenchymal cells (*ITGA7+*) [49]. While the authors discuss that some of this subclustering could be due to sample variability among male, female, injured, and uninjured samples, all five fibroblast types are evident in all samples. However, they suggest that conclusive evidence about the proportions of these subpopulations cannot yet be determined.

Other studies focused on identifying tendon progenitors, for example Still et al. and Guo et al. found seven and eight progenitor populations, respectively. Still et al. sequenced normal and tendinopathy human tendons, and specifically described mechanically responsive tendon progenitor populations (mrTPCs), pro-inflammatory TPCs (piTPCs), *SLC40A1+* TPCs, and highly clonogenic TPCs defined by high *NESTIN* expression [50]. However, it should be noted that these populations were determined from sequencing cells which were extracted from tendons, grown in culture, and subjected to a stimulatory mechanical force while in the culture system. Guo et al. harvested human tendinopathy tissue and cultured it for two passages before fluorescence activated cell sorting for tendon-derived stem cells (TDSCs) and scRNA-seq [51]. The eight identified populations were all a part of the tendon microenvironment and included pathways such as inflammation, migration, fibrosis, and ECM remodeling. The authors also specifically emphasized TDSC cluster 0, which was identified by overexpression of *AKR1C1* and *CFD*, and was the largest cell subset in damaged tendons. Overall, additional care should be taken when drawing conclusions from the sequencing of cells that were previously cultured, as the process of plating and culturing can change their gene expression [53,54].

Harvey et al. identified three tendon cell-related clusters: tenocytes, tendon fibro-adipoengic progenitors, and *Tppp3+* progenitor cells found in the tendon sheath [3]. Tenocytes were identified by the expression of *Fibromodulin* (*Fmod*), *Tnmd*, and *Thrombospondin 4* (*Thbs4*). Tendon fibro-adipogenic progenitors were defined by *Pdgfra* and *Ly6a* expression, and lack of *CD45*, *CD31*, and *Itga7*. Finally, the tendon stem cells were defined by expression of *Lubricin* (*Prg4*) and *Tppp3*. Further investigation of the *Tppp3+* stem cells through lineage tracing showed that they can proliferate and differentiate into tenocytes in response to tendon injury. These cells are also capable of self-renewal in the sheath.

As seen in these and other studies, there is not one unified classification system for tendon cells or their progenitors [3,48,49,50,51,52]. Continued use of advanced sequencing techniques such as scRNA-seq, as well as multiomics and spatial transcriptomics, will surely elucidate further information on tenocyte and fibroblast heterogeneity.

## 5. Tendon Heterogeneity across the Body

The size and shape of tendons can vary based on their location in the body; for example, the Achilles tendon is fibrous and cord-like, the rotator cuff tendons are flat and broad, and the flexor tendons of the hand are elongated and thin [55,56]. The structure of tendons shares many similarities with that of ligaments; both exhibit a general composition of dense connective tissue with type I and III collagen. However, there are distinct differences in cell content between ligaments and tendons; for example, ligaments comprise a greater proportion of collagen III and proteoglycan [57,58].

The function of a tendon also demonstrates variability, either being classified as energy storing or positional [13]. Energy-storing tendons, such as the Achilles and patellar tendons, can provide elasticity when they are extended (Table 1). In contrast, positional tendons, such as the supraspinatus tendon, are less elastic and remain stiff to transfer force to the bone [56,59,60]. These functional distinctions can be attributed to differences in cellular and ECM makeup. The former group of tendons has an increased proportion of glycosaminoglycan and water content compared to the latter group, resulting in a softer matrix and lower tissue stiffness level, enhancing its flexibility. One study proposed the increased elasticity seen in energy-storing tendons was due to lower interfascicular rigidity, facilitating the repeated sliding of fascicles during movement. Furthermore, the increased elastin content and trivalent cross-linking between collagen fibrils in energy-storing tendons appear to prevent mechanical disruption of the tendon during activities of high strain [61,62]. These properties thereby determine the biomechanics and locomotive capabilities of a tendon.

The diversity in protein profile and non-collagenous components is vast, even within tendons that share similar functions, such as the Achilles and patellar tendon [63,64]. On the transcriptional level, tendons also differ in the expression of genes involved in cell proliferation, extracellular matrix synthesis, and limb development. For example, the patellar and Achilles tendons have greater expression levels of collagen I synthesis gene *Col1a1* than the supraspinatus and flexor tendons [65]. Understanding the genetic and structural heterogeneity of tendons and the mechanisms that drive their variability is crucial to developing therapies for tendon healing.

## 6. Tendon Fibrosis

Tendons have a limited regenerative capacity, largely due to their low cellularity and vascularity [11]. Thus, tendon injuries will always heal with compromised mechanical and structural properties when compared to the original tissue [21,66]. The deposition of weaker, less-organized fibrotic scar tissue during healing also contributes to this change in tendon properties. As with skin fibrosis, tendon fibrosis occurs throughout the three stages of healing, with the main deposition of disorganized collagen and other ECM proteins occurring during the proliferative phase (Figure 1) [21]. The scar is also maintained long term in the remodeling phase. In tendons specifically, fibrosis is believed to be due to an imbalance of extrinsic pro-fibrotic fibroblast populations and intrinsic regenerative tenocytes [32,36].

If the fibrotic tissue forms between the tendon and surrounding tissue, it is referred to as an adhesion. Approximately 30% of tendon injuries lead to adhesions [15]. Adhesions are more likely to form after injury of intrasynovial tendons, such as the flexor tendons of the hands, compared to extrasynovial tendons, such as the Achilles and rotator cuff, due to the limited access of vasculature inside the synovium [5]. Once an adhesion forms, it limits the sliding motion of the tendon, thus impairing the range of motion for the affected joint [5]. Deep adhesions can cause pain in the tendon itself, as well as loss of function, and operative treatment to release adhesions may become necessary [67]. While this can be helpful in restoring range of motion, additional surgeries also create an increased risk for further fibrosis.

Current strategies for reducing adhesion formation include optimizing physical rehabilitation protocols; because tendons are shaped by the mechanical load at baseline, the mechanical load during healing plays a critical role [68]. Complete immobilization after tendon repair leads to more adhesions, while partial motion or passive motion reduces adhesions [5,69]. Other approaches have focused on suturing techniques and grafting of healthy tendons [14]. These techniques are useful but not sufficient to reduce or eliminate adhesions; therefore, innovative solutions are being explored, which combine existing techniques with biomaterials and biological anti-fibrotic targets.

## 7. Biomaterials and Novel Solutions to Tendon Fibrosis

### 7.1. Overview

The pervasiveness of tendon fibrosis has created an unmet need for novel tissue engineering techniques. Currently, there are many approaches to create tendon regeneration strategies, such as tissue-engineered scaffolds, injectable drug-loaded hydrogels, and anti-adhesive biomaterials (Figure 3). These tissue-engineered solutions comprise combinations of various biomaterials, drugs, growth factors, and stem or differentiated cells [70]. Careful choice or exclusion of each of these components is critical to identifying an optimal solution. For the purposes of this discussion related to tendon fibrosis, we will briefly discuss scaffolds and hydrogels and then focus on anti-adhesive materials.

The primary goal of tissue-engineered tendon repair scaffolds is to create a biocompatible solution which has good mechanical properties, can be incorporated into existing tissue, and reduces fibrosis [71]. Scaffolds can be made of natural or synthetic biomaterials and can be loaded with growth factors or other drugs to promote tendon healing [19,72]. One study used a poly-lactic-co-glycolic acid (PLGA) backbone layered with a heparin/fibrin-based delivery system and contained both adipose-derived mesenchymal stem cells and platelet-derived growth factor-BB (PDGF-BB). The study confirmed cell viability and drug delivery in vivo and is a great example of the way biomaterials can be combined with cells and growth factors to target tendon regeneration [73]. In terms of injectable treatments, hydrogels are widely used because they are biodegradable and resemble the physical properties of biological tissue, such that they do not interfere but still deliver the drug of interest. Examples of hydrogels used include hyaluronic acid and bioactive glass/sodium alginate, and delivery agents include fibromodulin and Mg^2+^ with curcumin [74,75,76].

### 7.2. Anti-Adhesive Biomaterials

Several anti-adhesive biomaterials have been developed, which can be used as a physical barrier between tendons and the surrounding tissue after repair. These materials promote intrinsic tendon healing, deliver drugs of interest, and allow for smooth gliding between tendons and the surrounding tissue [77]. There are several categories of biomaterials used for anti-adhesive strategies: hydrogels, electrospun fiber membranes, and absorbable films [78].

Hydrogels are beneficial because they are made up of polymers, which can be modulated to control mechanical properties and biochemical degradation. One example, Seprafilm, is a sodium hyaluronic acid and carboxymethylcellulose hydrogel, which has been shown to significantly decrease adhesion formation compared to the control, where only sutures were placed [79,80]. However, in general, Seprafilm and other technologies still do not fully eliminate fibrosis, and better solutions are still being explored. Hydrogels are also immensely useful because they can be loaded with tendon growth factors or anti-fibrotic drugs such as 5-fluorouracil or corticosteroid [81,82]. However, the dosing of these drugs must be carefully determined, or adhesions may instead increase. Another drawback of hydrogels is that they may be degraded or washed out more easily than other options, resulting in lower efficacy [78].

Electrospun fiber membranes are created by spinning a polymer using electric fields to create a nano-sized continuous fiber [83]. They can be made of materials such as poly(l-lactic acid) (PLLA), silk, and poly(ε-caprolactone) (PCL) and provide an advantage over hydrogels by taking longer to degrade in vivo [78]. They also demonstrate good biocompatibility and have limited immune reactivity. However, electrospun fiber membranes are more difficult to place than hydrogels, can cause an increase in tendon thickness, and can be more costly and complicated to manufacture. For example, the electric field used to create the fibers can disrupt the therapeutic activity of associated drugs [78].

Absorbable films can be made from either natural biological tissue, such as a decellularized amnion or porcine peritoneum, or synthetic polymer films [77,84,85]. Benefits include biocompatibility and biodegradability, as well as immune reaction reduction. One drawback to absorbable films is that they are more technically difficult to maneuver and place, especially in situations of complex lesions to the tendon [78]. There is also still work to be performed regarding optimization of their degradation rate and mechanical properties [78].

There are many other examples of tissue engineering solutions for tendon regeneration which are too numerous to discuss in detail here. Additional examples of materials, cells, and growth factors or drugs which are used in tissue engineering are discussed in Figure 3. Continued study of these tissue engineering therapies will lead to improvement of tendon healing and reduction in fibrosis.

## 8. Future Directions

There remain ample opportunities for improving tendon regeneration and reducing fibrosis. Some of the key gaps in the field include better identification of tenocyte markers, understanding of the contribution of tendon progenitor populations to tendon healing, and elucidation of therapeutic targets for reducing fibrosis. An increasing number of studies using sequencing technologies are being published to further understand tendon cell populations. Among these are bulk RNA sequencing and scRNA-seq, which provide data on the transcriptome of whole tissue and the increased resolution on cell diversity within tissue, respectively [86,87]. scRNA-seq can also be paired with assay for transposase-accessible chromatin with sequencing (ATAC-seq) to collect both epigenetic and transcriptomic information (also known as multiomics), further deepening our understanding [49]. Finally, spatial studies such as transcriptomics allow a detailed identification of cell populations’ spatial relationships to each other [51,88]. A new spatial proteomics technique called co-detection by indexing (CODEX) is being expanded into multiple tissues and can provide highly multiplexed spatial transcriptomic information [89]. Combining the continued advancement of the above-mentioned sequencing techniques with other tools such as lineage tracing and immunohistochemistry will help us better understand the key cell populations and how to target them.

A better investigation of the immune system’s role in tendon healing would also greatly advance the field. Current work exploring the relationship between the M1 and M2 phenotypes of macrophages and tendon healing is one avenue of exploring the role of the immune system. Although a simplified explanation, M1 macrophages are generally thought to be pro-inflammatory and shown to play a role in the inflammatory phase of tendon healing, while M2 macrophages are pro-regenerative and promote fibroblast proliferation to repair the tendon [90]. A better understanding of not just macrophages but the wider role of the immune system on tendon healing may provide novel targets for improved healing.

With regards to creating systems to study tenocytes and tendon injury more effectively, tenocytes are known to be difficult to maintain in in vitro culture, which means that drawing conclusions from in vitro studies alone is difficult. Advancements in tenocyte culture methods, for example creating better 3D culture systems which replicate in vivo conditions, would provide additional means to study cells of interest [90]. Of course, any tissue-engineered solution would present some drawbacks, and explant culture systems that are being developed also present a promising way to study tendons in vitro [91]. These systems involve placing whole tissues directly into culture for a better replication of their natural environment during experimentation [91]. Beyond in vitro studies, conducting additional studies in both small and large animal studies will identify targets for eventual clinical trials. Bridging the gap between in vitro, in vivo, and clinical trials will lead to the ultimate goal of translation of these techniques to humans and prevention of tendon fibrosis.

## 9. Conclusions

Despite the vast clinical burden of tendon injuries, there remains a limited understanding of therapeutic targets for tendon fibrosis. Both intrinsic tenocytes and extrinsic fibroblasts play a role in healing, and fibrosis is caused by an imbalance between the intrinsic regenerative cells and extrinsic fibrotic cells. This fibrosis can be extremely debilitating for patients, causing pain and limited range of motion. To address these clinical concerns, there are a growing number of studies using sequencing technologies to identify tendon fibroblast heterogeneity. However, there is still much to be explored, including multiomics, spatial transcriptomics, and CODEX. To implement anti-fibrotic treatments once targets are identified, optimal scaffold, drug delivery, and anti-adhesive tissue-engineered solutions must be developed. A better understanding of cell subpopulations which play a role in tendon healing and fibrosis will provide novel therapeutic targets to reduce the clinical burden of tendon fibrosis. 

## Figures and Tables

**Figure 1 biomedicines-12-00859-f001:**
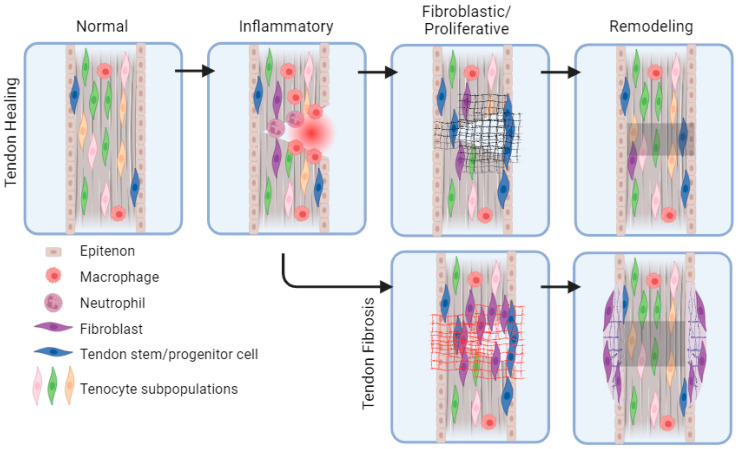
Schematic of stages of tendon healing and fibrosis and the cells involved. A normal tendon is made up of multiple populations of tenocytes and a tendon stem and progenitor cells in a collagen matrix and is surrounded by the epitenon. Macrophages have also been documented to be present at baseline. After injury, tendon healing and fibrosis go through three overlapping phases: inflammatory, fibrotic/proliferative, and remodeling. In normal tendon healing, macrophages, neutrophils, and fibroblasts migrate into the site of the injury during the inflammatory phase. Then, intrinsic tenocytes and extrinsic fibroblasts begin laying down collagen III in the proliferative phase. In the remodeling phase, cellularity decreases, and collagen III is replaced with collagen I. All postnatal tendons heal with a scar, which has a compromised structure compared to normal tendons; however, some tendons go through a process of “over-healing” following the inflammatory phase (bottom row). In these fibrotic tendons, fibroblasts lay down an excess of disorganized collagen and other matrix proteins during the proliferative phase. During the remodeling phase, the increased scar is maintained by replacing collagen III with collagen I. Figure created using BioRender.com (accessed on 8 March 2024).

**Figure 2 biomedicines-12-00859-f002:**
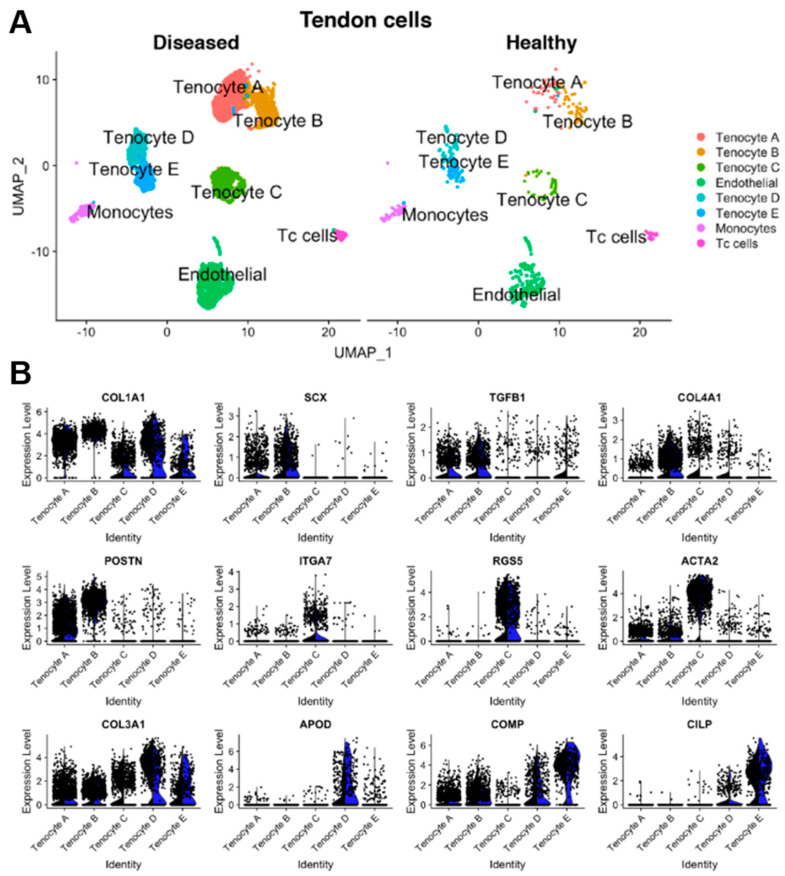
Representative image of tenocyte subpopulations from Kendal et al. [49]. (**A**) Uniform Manifold Approximation and Projection (UMAP) showing diseased, or tendinopathy, and healthy patient samples. Eight overall cell populations and five tenocyte populations were identified. All cell populations were present in both diseased and healthy tendon tissue. Tenocytes were defined as cells expressing *COL1A1* or *COL1A2*. (**B**) Split Violin plots displaying gene expression for diseased (black) versus healthy (blue) tenocytes in all five subpopulations. Every dot represents an individual cell’s gene expression level. Figure adapted from Kendal et al. [49], an open access publication.

**Figure 3 biomedicines-12-00859-f003:**
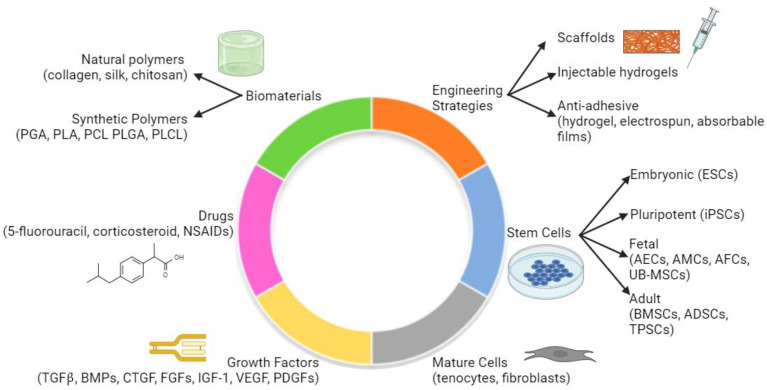
Considerations for tendon tissue engineering. Tendon tissue engineering is dependent on multiple factors and combinations of these factors: biomaterials, engineering strategies, choice of stem cells or mature cells, growth factors, and drugs. Abbreviations. PGA: polyglycolic acids, PLA: polylactic acids, PCL: polycaprolactones, PLGA: poly(lactic-co-glycolic) acids, PLCL: poly (lactil-co-captolactone) acids, ESCs: embryonic stem cells, iPSCs: induced pluripotent stem cells, AECs: amniotic epithelial stem cells, AMCs: amniotic mesenchymal stem cells, AFCs: amniotic fluid stem cells, UB-MSCs: umbilical cord mesenchymal stem cells, BMSCs: bone marrow mesenchymal stem cells, ADSCs: adipose derived mesenchymal stem cells, TPSCs: tendon progenitors stem cells, TGFβ: transforming growth factor beta, BMPs: bone morphogenetic proteins, CTGF: connective tissue growth factor, FGFs: fibroblastic growth factors, IGF-1: insulin-like growth factor-1, VEGF: vascular endothelial growth factor, PDGFs: platelet-derived growth factor, NSAIDs: non-steroidal anti-inflammatory drugs. Figure adapted from Citeroni et al. [70], an open access publication. Adaptation created using BioRender.com.

**Table 2 biomedicines-12-00859-t002:** Key cells and growth factors involved in tendon healing.

Key Players in Tendon Healing			
Cell Type	Primary Phase Present	Primary Function	Reference
Erythrocytes	Early inflammatory	Can be present from broken vessels	Connizzo et al. [18]
Platelets	Early inflammatory	Clot blood vessels broke in injury, secrete PDGF	Chen et al. [22]
Neutrophils	First 24 h	Phagocytose necrotic tissue	Marsolais et al. [20]
Macrophages	Inflammatory phase, typically after neutrophils	Phagocytosis; release of growth factors to stimulate fibroblasts, increase ECM synthesis, and decrease ECM degradation	Marsolais et al. [20]
Fibroblasts	Inflammatory	Migrate into site of injury	Lomas et al. [19]; Parker et al. [16]
	Proliferative	Deposit collagen III	
	Remodeling	Deposit collagen I	
Tenocytes	Present in tendon for all phases	Contribute to intrinsic tendon healing and collagen formation	Nichols et al. [21]
Tendon Stem and Progenitor Cells	Present in sheath for all phases	Differentiate to form mature tenocytes for tissue regeneration	Harvey et al. [3]
**Growth Factor or Enzyme**	**Primary Phase Present**	**Primary Function**	**Reference**
PDGF	Inflammatory and remodeling	Induces synthesis of IGF-I and IGF receptors; stimulate collagen, protein, and DNA production; stimulate macrophage migration	Chen et al. [22]
IGF-I	Inflammatory and proliferative	Stimulate ECM formation; stimulates migration and proliferation of fibroblasts and tendon stem cells	Miescher et al. [23]
VEGF	Proliferative and remodeling	Promotes angiogenesis	Lin et al. [24]
TGF-β	All phases	Many functions including collagen production, cell viability, promoting *Scx* expression	Li et al. [25]
bFGF	All phases	Angiogenesis, cellular migration and proliferation; Promotes expression of other growth factors	Lu et al. [26]
MMPs	All phases	Degradation and turnover of collagen, glycoproteins, proteoglycans	Andarawis-Puri et al. [7]

Note that many cells and growth factors are present for multiple phases of healing, but the predominant phase(s) have been listed in this table. Abbreviations. ECM: extracellular matrix, PDGF: platelet-derived growth factor, IGF-I: insulin-like growth factor-I, VEGF: vascular endothelial growth factor, TGF-β: transforming growth factor-beta, *Scx*: *scleraxis*, bFGF: basic fibroblast growth factor, MMPs: matrix metalloproteinases.

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
