# Peer review of "Understanding Tendon Fibroblast Biology and Heterogeneity"

_biomedicines, 2024, doi:10.3390/biomedicines12040859_

Round 1

Reviewer 1 Report

Comments and Suggestions for Authors

Tendon damage is one of the factors that can have an impact on human life quality because of its low regenerating ability and the creation of scar tissue at the site of damage. The techniques of treatment available to physicians do not always have the desired therapeutic impact, necessitating the quest for alternative means of treatment, including the use of cellular technology. This, in turn, necessitates a clear understanding of the mechanisms of tendon regeneration, as well as the role of cells and biologically active compounds in this process, including their own tendon cells and those attracted from the outside, and the possibility of influencing them with therapeutic technologies. As a result, this review piece is absolutely relevant.

According to PubMed, there are numerous types of reviews that address both the processes of tendon regeneration and how to therapeutically boost this process.
In particular:

Squier K, Mousavizadeh R, Damji F, Beck C, Hunt M, Scott A. In vitro collagen biomarkers in mechanically stimulated human tendon cells: a systematic review. Connect Tissue Res. 2024 Feb 20:1-13. doi: 10.1080/03008207.2024.2313582.

He P, Ruan D, Huang Z, Wang C, Xu Y, Cai H, Liu H, Fei Y, Heng BC, Chen W, Shen W. Comparison of Tendon Development Versus Tendon Healing and Regeneration. Front Cell Dev Biol. 2022 Jan 24;10:821667. doi: 10.3389/fcell.2022.821667.

Nichols AEC, Best KT, Loiselle AE. The cellular basis of fibrotic tendon healing: challenges and opportunities. Transl Res. 2019 Jul;209:156-168. doi: 10.1016/j.trsl.2019.02.002.

In their review, the authors attempted to focus data on the mechanisms of the regenerative process in tendons, with a particular emphasis on the heterogeneity of tendon fibroblasts and the need to study their role in the process of starting fibrosis, looking for potential points of influence on the interruption of scar formation.

The review includes 50% of articles under 5 years old that are relevant to drafting an article. There was no evidence of self-citation.

In the review, the authors discuss the variety of tendon fibroblasts, the involvement of various biologically active chemicals in the process of tendon regeneration, and innovative therapeutic techniques for the treatment of tendon injuries, citing literary sources.

There is no doubt about the importance of drawings and tables in the work, since they contain accurate data and help to the visual sense of the content.

One disadvantage of the work is the absence of a description of the literature search method (PubMed, Scopus, Web of Science electronic databases), keywords, analysis period (years), and data analysis criteria. 2) Place the note to Table 1 under the table, not in the title (lines 86-90). 3) Abbreviate differentiation clusters in capital letters (line 215). 4) The paper does not discuss the role of matrix metalloproteases.

Author Response

Reviewer 1

Tendon damage is one of the factors that can have an impact on human life quality because of its low regenerating ability and the creation of scar tissue at the site of damage. The techniques of treatment available to physicians do not always have the desired therapeutic impact, necessitating the quest for alternative means of treatment, including the use of cellular technology. This, in turn, necessitates a clear understanding of the mechanisms of tendon regeneration, as well as the role of cells and biologically active compounds in this process, including their own tendon cells and those attracted from the outside, and the possibility of influencing them with therapeutic technologies. As a result, this review piece is absolutely relevant.

According to PubMed, there are numerous types of reviews that address both the processes of tendon regeneration and how to therapeutically boost this process.

In particular:

Squier K, Mousavizadeh R, Damji F, Beck C, Hunt M, Scott A. In vitro collagen biomarkers in mechanically stimulated human tendon cells: a systematic review. Connect Tissue Res. 2024 Feb 20:1-13. doi: 10.1080/03008207.2024.2313582.

He P, Ruan D, Huang Z, Wang C, Xu Y, Cai H, Liu H, Fei Y, Heng BC, Chen W, Shen W. Comparison of Tendon Development Versus Tendon Healing and Regeneration. Front Cell Dev Biol. 2022 Jan 24;10:821667. doi: 10.3389/fcell.2022.821667.

Nichols AEC, Best KT, Loiselle AE. The cellular basis of fibrotic tendon healing: challenges and opportunities. Transl Res. 2019 Jul;209:156-168. doi: 10.1016/j.trsl.2019.02.002.

In their review, the authors attempted to focus data on the mechanisms of the regenerative process in tendons, with a particular emphasis on the heterogeneity of tendon fibroblasts and the need to study their role in the process of starting fibrosis, looking for potential points of influence on the interruption of scar formation.

The review includes 50% of articles under 5 years old that are relevant to drafting an article. There was no evidence of self-citation.

In the review, the authors discuss the variety of tendon fibroblasts, the involvement of various biologically active chemicals in the process of tendon regeneration, and innovative therapeutic techniques for the treatment of tendon injuries, citing literary sources.

There is no doubt about the importance of drawings and tables in the work, since they contain accurate data and help to the visual sense of the content.

We thank the reviewer for the positive appraisal of our manuscript.

1) One disadvantage of the work is the absence of a description of the literature search method (PubMed, Scopus, Web of Science electronic databases), keywords, analysis period (years), and data analysis criteria.

Thank you for your comment. This was not intended as a systematic review, and thus we did not provide a detailed methods statement. However, we have now included a sentence on page 3 at the end of the Introduction to include the search terms we used on PubMed:

We conducted this review using PubMed with search terms “tendon fibrosis,” “tendon heterogeneity,” “skin fibrosis,” and mainly focused on papers published in the last five years.

2) Place the note to Table 1 under the table, not in the title (lines 86-90).

This has been corrected.

3) Abbreviate differentiation clusters in capital letters (line 215).

This has been changed as requested.

4) The paper does not discuss the role of matrix metalloproteases.

Thank you for your comment. A paragraph has been added on page 5 to the end of section 2, Overview of Tendon Healing: Cells and Signals, to discuss matrix metalloproteinases. MMPs have also been added to Table 2.

Matrix metalloproteinases (MMPs) are a family of enzymes which are another important factor in all phases of tendon healing [7]. In normal tendon development and maintenance, MMPs process and turn over collagen and promote new fibril growth and assembly. After injury, some MMPs degrade collagens, while others degrade glycoproteins and proteoglycans present in the tendons [14]. This degradation creates space for new, healthy tendon components to be produced and remodeled [14]. However, long-term activation of MMPs can be detrimental, because MMPs can create a low level of consistent inflammation which can weaken tendons and contribute to tendinopathy or rupture [29,30].

Addition to Table 2 (p20):

MMPs

All phases

Degradation and turnover of collagen, glycoproteins, proteoglycans

Andarawis-Puri et al. [7]

Reviewer 2 Report

Comments and Suggestions for Authors

This review devoted to heterogeneity of tendon fibroblasts on the cellular level seems useful from the point of view of therapeutic strategies for addressing tendon fibrosis. This review focusing on the heterogeneity of tendon fibroblasts at the cellular level appears to be useful in terms of therapeutic strategies for the treatment of tendon fibrosis.

The review is well structured and the literature has been selected.

However, I recommend that authors add figures from the most significant cited articles to the section

Author Response

Reviewer 2

This review devoted to heterogeneity of tendon fibroblasts on the cellular level seems useful from the point of view of therapeutic strategies for addressing tendon fibrosis. This review focusing on the heterogeneity of tendon fibroblasts at the cellular level appears to be useful in terms of therapeutic strategies for the treatment of tendon fibrosis.

The review is well structured and the literature has been selected.

  • However, I recommend that authors add figures from the most significant cited articles to the section

Thank you for the comment. We have added Figure 2 (p22-23) to section 4.2., Fibroblast heterogeneity in tendon, to summarize critical findings of Kendal et al. We believe this to be representative of the type of scRNA-seq analysis used to look for tenocyte subpopulations and tenocyte heterogeneity. Figure 2 is shown below.

Figure 2. Representative image of tenocyte subpopulations from Kendal et al.

(A) Uniform Manifold Approximation and Projection (UMAP) showing diseased, or tendinopathy, and healthy patient samples. Eight overall cell populations and five tenocyte populations were identified. All cell populations were present in both diseased and healthy tendon tissue. Tenocytes were defined as cells expressing COL1A1 or COL1A2. (B) Split Violin plots displaying gene expression for diseased (black) versus healthy (blue) tenocytes in all five subpopulations. Every dot represents an individual cell’s gene expression level. Figure adapted from Kendal et al. [48], an open access publication.

Reviewer 3 Report

Comments and Suggestions for Authors

The authors have made an interesting summary of the understanding tendon fibroblast biology and heterogeneity, but some improvements should be made:

  1. The title should be “Understanding tendon fibroblasts biology and heterogeneity.”
  2. The references are not written according to the recommendation of the Biomedicine journal, the number should be written not superscript but in square brackets.
  3. I recommend a figure in the introduction that reflects the types of tendons and their characteristics.
  4.  Line 73 In Fig 1, healing should be written without capital letter and after the title of the figure should be a space (ENTER) delimitating the figure and the text that follows and that explains the figure.
  5. Chapter 4 should be “Heterogeneity in Fibroblasts Function” and 4.1. Fibroblasts heterogeneity in skin.
  6. Line 163, please explain the acronym GO-term analysis.
  7. For the chapter 5. Tendon Heterogeneity Across the Body a table will be more than appropriate to be done in order to highlight the different tendons.
  8. For tendon fibrosis some figures are necessary in order to visualize better the process
  9. Line 311 PDGF-BB what does it represent?
  10. Line 317 at Mg 2+, 2+ should be written with superscript.
  11. The subchapter 7.2. Anti-Adhesive Biomaterials it is very short, so it should be developed, and a table should be shown in which the advantages for each biomaterial (electrospun fiber, hydrogels, absorbable films) described should be highlighted.
  12. Line 343-344 should be reformulated.
  13. Line 353 define ATAC seq. 
  14. Line 376 please explain the concept “explant culture systems.”
  15. Line 390 before anti-fibrotic there are more than one space.

Author Response

Reviewer 3

The authors have made an interesting summary of the understanding tendon fibroblast biology and heterogeneity, but some improvements should be made:

  • The title should be “Understanding tendon fibroblasts biology and heterogeneity.”

Thank you for the comment. This has been changed as requested.

  • The references are not written according to the recommendation of the Biomedicine journal, the number should be written not superscript but in square brackets.

Thank you for the comment. This has been corrected such that the citation format is that of Biomedicines.

  • I recommend a figure in the introduction that reflects the types of tendons and their characteristics.

Thank you for the suggestion. We have added Table 1 (p19) as a summary of the types of tendons and their characteristics:

Type of Tendon

Definition/Function

Key Examples

Synovial

Surrounded by synovial sheath and bathed in synovial fluid

Flexor tendons of hands and feet

Non-synovial

Surrounded by paratenon

Achilles, Rotator cuff tendons

Energy-storing

Provide elasticity

Achilles, Patellar

Positional

Remain stiff to transfer force to bone

Supraspinatus

Table 1. Summary of Types of Tendon with Examples.

There are two main classification systems for tendons: synovial or non-synovial, and energy storing or positional. Definitions and examples are listed for each. This table was summarized from Kaya et al. [56].

  • Line 73 In Fig 1, healing should be written without capital letter and after the title of the figure should be a space (ENTER) delimitating the figure and the text that follows and that explains the figure.

This formatting change has been made for all figures as requested.

  • Chapter 4 should be “Heterogeneity in Fibroblasts Function” and 4.1. Fibroblasts heterogeneity in skin.

Based on this comment, all chapter subtitles have been changed to sentence case capitalization pattern.

  • Line 163, please explain the acronym GO-term analysis.

This has been changed as requested.

  • the chapter 5. Tendon Heterogeneity Across the Body a table will be more than appropriate to be done in order to highlight the different tendons.

Thank you for this comment. Table 1, which was added in response to comment 3 above, addresses types of tendons and their classification, as well as key examples. We have referenced Table 1 in chapter 5.

  • For tendon fibrosis some figures are necessary in order to visualize better the process

Thank you for your comment and we agree that a figure helps with understanding of tendon fibrosis. The progression of tendon fibrosis and cells involved is addressed in Figure 1.

  • Line 311 PDGF-BB what does it represent?

The complete spelling of this term has been added to line 311.

  • Line 317 at Mg 2+, 2+ should be written with superscript.

This has been corrected as requested.

  • The subchapter 7.2. Anti-Adhesive Biomaterials it is very short, so it should be developed, and a table should be shown in which the advantages for each biomaterial (electrospun fiber, hydrogels, absorbable films) described should be highlighted.

Thank you for the comment. We agree with you and have added to the text in this section. We have now included benefits and drawbacks for each of the 3 biomaterial methods described in subchapter 7.2. These changes include the following:

Page 14: Hydrogels are beneficial because they are made up of polymers, which can be modulated to control mechanical properties and biochemical degradation.

Page 14-15: Another drawback of hydrogels is that they may be degraded or washed out more easily than other options, resulting in lower efficacy [79].

Page 15: They also demonstrate good biocompatibility and have limited immune reactivity. However, electrospun fiber membranes are more difficult to place than hydrogels, can cause an increase in tendon thickness, and can be more costly and complicated to manufacture. For example, the electric field used to create the fibers can disrupt the therapeutic activity of associated drugs [79].

Page 15: Benefits include biocompatibility and biodegradability, as well as immune reaction reduction.

Page 15: There is also still work to be done regarding optimization of their degradation rate and mechanical properties [79].

Page 15: Continued study of these tissue engineering therapies will lead to improvement of tendon healing and reduction of fibrosis.

  • Line 343-344 should be reformulated.

Thank you for this point. This line has been reworded as requested.

  • Line 353 define ATAC seq.

This definition has been added as requested.

  • Line 376 please explain the concept “explant culture systems.”

A sentence has been added to better explain the concept and function of explant culture systems:

These systems involve placing whole tissues directly into culture for a better replication of their natural environment during experimentation [92].

  • Line 390 before anti-fibrotic there are more than one space.

This has been corrected as requested.

Round 2

Reviewer 3 Report

Comments and Suggestions for Authors

Authors made the proper modifications according to the recommandations.